



# Air pollution emission inventory using national high-resolution spatial parameters for the Nordic countries

Ville-Veikko Paunu[1], Niko Karvosenoja[1], David Segersson[2], Susana López-Aparicio[3], Ole-Kenneth Nielsen[4], Marlene Schmidt Plejdrup[4], Throstur Thorsteinsson[5], Dam Thanh Vo[3], Jeroen Kuenen[6], Hugo
Denier van der Gon[6], Jukka-Pekka Jalkanen[7], Jørgen Brandt[4,8], Camilla Geels[4]

[1] Finnish Environment Institute (Syke), Latokartononkaari 11, FI-00790 Helsinki, Finland
[2] Swedish Meteorological and Hydrological Institute, SMHI, Folkborgsvägen 1, SE-601 76 Norrköping, Sweden
[3] NILU – Norwegian Institute for Air Research, Instituttveien 18, NO-2007 Kjeller, Norway
[4] Department of Environmental Science, Aarhus University, Frederiksborgvej 399, P.O.Box 358, 4000 Roskilde, Denmark
[5] Environment and Natural Resources & Institute of Earth Sciences, University of Iceland, Sturlugata 7, 101 Reykjavik, Iceland
[6] Department of Climate, Air and Sustainability, TNO, Utrecht, The Netherlands
[7] Finnish Meteorological Institute, Atmospheric Composition Research, P.O. Box 503, FI-00101 Helsinki, Finland
[8] iClimate – Aarhus University interdisciplinary Centre for Climate Change, Frederiksborgvej 399, P.O.Box 358, 4000
Roskilde, Denmark

*Correspondence to*: Ville-Veikko Paunu (ville-veikko.paunu@syke.fi)

**Abstract.** Air pollution is an important cause of adverse health effects even in the Nordic countries, which have relatively good air quality. Modelling-based air quality assessment of the health impacts relies on reliable model estimates of ambient air pollution concentrations, which furthermore rely on good quality spatially resolved emission data. While quantitative
emission estimates are the cornerstone of good emission data, description of the spatial distribution of the emissions is especially important for local air quality modelling on high resolution. In this paper we present a new air pollution emission inventory for the Nordic countries with high-resolution spatial allocation (1 km × 1 km) covering years 1990, 1995, 2000, 2005, 2010, 2012, and 2014. The inventory is available at https://doi.org/10.5281/ZENODO.8070155 (Paunu et al., 2023)). To study the impact of applying national data and methods to the spatial distribution of the emissions, we compared road
transport and machinery and off-road sectors to CAMS-REGv4.2, which used consistent spatial distribution method throughout Europe for each sector. Road transport is a sector with well-established proxies for spatial distribution, while for machinery and off-road the choice of proxies is not as straightforward as it includes a variety of different type of vehicles and machines operating in various environments. We found that the CAMS-REGv4.2 was able to produce similar spatial patterns as our Nordic inventory for the selected sectors. However, the resolution of our Nordic inventory allows for more detailed
impact assessment than the CAMS-REGv4.2, which had resolution of 0.1° × 0.05° (longitude-latitude, roughly 5.5 km × 3.5-6.5 km in the Nordic countries). The EMEP/EEA Guidebook chapter on spatial mapping of emissions has recommendations for the sectoral proxies. Based on our analysis we argue that the guidebook should have separate recommendations for proxies for several sub-categories of the machinery and off-road sectors, instead of including them within broader sectors. We suggest that land use data is the best starting point for proxies for many of the subsectors, and they can be combined with other suitable
data to enhance the spatial distribution. For road transport, measured traffic flow data should be utilized where possible, to support modelled data in the proxies.

## 1. Introduction

Air pollution persists as one of the major environmental challenges in the world (Landrigan et al., 2018). Even in the Nordic countries (Denmark, Finland, Iceland, Norway, and Sweden), where the air is considered relatively clean, air pollution causes
adverse health effects (Brandt et al., 2013; European Environment Agency (EEA), 2021; Geels et al., 2021; Lehtomäki et al., 2020).



While transboundary air pollution contributes heavily to air quality in the Nordic countries (Im et al., 2019), emissions from low emission height sources often have large impact on local air quality, especially in the urban areas (Kukkonen et al., 2020; Pisoni et al., 2022). Therefore, when calculating population exposure, the correct representation of local emissions on national

or European scale emission inventories is crucial. There are two important aspects of this: quantitative emission estimates for different emission sources and their spatial distribution. The quality of emission estimates depends on the uncertainties of activity data and emission factors for the different emission sources (European Environment Agency, 2019). For spatial distribution, the ability to represent the location of the emissions is not the only essential issue, but also the resolution with which the assessment is carried out is important (Karvosenoja et al., 2011; Korhonen et al., 2019; Maes et al., 2009; Schaap et

al., 2015; Thunis, 2018; Zheng et al., 2017). This study focuses on the spatial distribution of the emissions.

There are few emission sources that can be described with actual, specific location. Large power and industrial plants have known locations, often significant emissions, and can be included in the inventories as point sources with exact coordinates. There are also special cases such as marine traffic, which can be located using tracking systems of actual movement and technical details of the source, e.g. AIS (Automatic Identification System) data (Jalkanen et al., 2009, 2012; Johansson et al.,

2017). However, source locations in many cases need to be described using proxies, as the data on the locations of individual sources might not be available, or they don't include enough relevant technical details or other information to estimate the emissions for each source.

Commonly used data for proxies include land cover, road network and population density data (Geng et al., 2017; Trombetti et al., 2018). Other data sources include satellite observations, e.g. for flaring or wildfire emissions, where the emission

calculation can also be based on the same data (Böttcher et al., 2021; McCarty et al., 2012). Other, less common, data are also used, such as night-time lights (Zheng et al., 2017). The distribution can be based on single attribute, such as area of the land cover class. Often several spatial datasets are combined to create the proxy, and other data can be used to develop the proxy further. For example, agricultural areas, such as fields, can be combined with animal numbers to distribute emissions from agriculture (Plejdrup et al., 2018). The EMEP/EEA Guidebook chapter on spatial mapping of emissions (European

Environment Agency, 2019) proposes three quality levels for spatial proxies for each sector (Tier 3 being most closely related to the spatial activity, and Tier 1 the simplest method, the latter only to be used for sources which are relatively unimportant for overall emissions).

Some sectors have more established spatial distribution methods than others. Road transport is an example of such a sector. Road network data, either open source (such as OpenStreetMap) or commercially available, is often used as the basis for the

proxy (Kuenen et al., 2022; Li et al., 2017; Romero et al., 2020; Trombetti et al., 2018). Road and street classes can be used to weight the activity on different road types (Zheng et al., 2014). Additional data on, e.g., traffic volumes, either measured or based on traffic models, average driving speed, driving patterns or information on the vehicle fleet composition (Jensen et al., 2008, 2019) allow more detailed weighting. These can help with creating separate proxies for exhaust and dust (resuspension) emissions. The proxy can also be based on bottom-up emission calculation (Andersson et al., 2019; Grythe et al., 2022), i.e.,

the emissions are calculated on street or even single vehicle level and then aggregated to a grid. It is also possible to calculate the emissions (and, consequently, their spatial distribution) completely by bottom-up methods (Breuer et al., 2020, 2021). For road transport, the EMEP/EEA Guidebook recommends traffic flows, separately per vehicle type, as Tier 3 level proxy.

Mobile machinery and off-road mobile sources is a sector with more complex spatial distribution question as it includes a vast amount of different types of vehicles and machines that operate in various environments. For example, railways, air traffic

landing and take-off (LTO) cycles and machinery used in agriculture, industry, construction, or maintenance all need very different data to represent the activity spatially. For some of the activities the choice of proxy data is relatively straightforward. For example, railway emissions can be distributed to non-electrified railway network, or air traffic LTO to airports (and their surroundings). However, for some other types of machinery, e.g., the ones used in construction, it is more difficult, as the locations of construction sites change regularly, and GIS data of such activities might not be easily available in consistent



forms. A proxy that would represent the activity over several years is hard to develop and would require detailed information on the construction activity over time, and data that define the energy used associated with the different phases of the construction project (Lopez-Aparicio and Grythe, 2022).

For different types of machinery, population density is often used as a proxy (Geng et al., 2017), sometimes separated between urban and rural population to cover the differences between those area types (Zheng et al., 2017). However, population have

been shown to be inadequate proxy data for most of the off-road and machinery emissions (Karvosenoja et al., 2020). In general, land use data is often a good starting point for the proxy (for example fields for agricultural machinery and industrial areas for industrial machinery). Other options include night lights (Zheng et al., 2017) or gross domestic product (Li et al., 2017).

In the EMEP/EEA Guidebook (European Environment Agency, 2019), recommendations for spatial proxies for machinery

and off-road sectors are divided between several sectors. Some have their distinct recommendations, such as aviation (airport locations combined with take-off and landing statistics or detailed estimation of emissions by aircraft) and railways (rail network weighted with population or, preferably, diesel rail traffic). Others are combined with stationary combustion (manufacturing industries and construction, commercial, residential, agriculture, other). For all of these, Tier 1 recommendation is land cover data (or population for manufacturing industries and construction). For industrial and

commercial machinery the guidebook Tier 3 recommendation is related point source data. For residential (household and gardening) and agriculture/forestry, Tier 3 is detailed fuel deliveries accompanied with modelled fuel use estimates based on "population density and/or household numbers and types" and employment data, respectively.

Emissions from the Nordic countries have been assessed nationally and reported as a part of continental scale emission inventories. The Nordic countries report their official emission inventories according to the United Nations Economic

Commission for Europe (UNECE) Convention on Long-range Transboundary Air Pollution (CLRTAP). These inventories are reported as national totals annually and as gridded emissions every four years on a $0.1° \times 0.1°$ resolution (approx. 4-6 km (east-west) $\times$ 11 km (north-south) for the Nordic area).

The emissions of the Nordic countries are also included in several independent emission inventories. The main examples of global inventories which include spatial distributions of the emissions are The Greenhouse gas - Air pollution Interactions and

Synergies (GAINS) model (Klimont et al., 2017) and Emissions Database for Global Atmospheric Research (EDGAR) (Crippa et al., 2018, 2019). Important inventories with regional and global coverage are the CAMS regional and global inventories (CAMS-REG (Kuenen et al., 2022) and CAMS-GLOB-ANT (Doumbia et al., 2021)). Since all these inventories are for either the European or global domain, the spatial distribution is typically based on a more generic methodology which does not take specific Nordic characteristics into account. In the emission estimates themselves however, these can and have been considered

as some of these inventories are either build on the national reported emission inventories (CAMS), or actively compared to them (GAINS). The inventories have been used extensively in air pollution impact assessments. The GAMS-REG-inventory (and its predecessor TNO MACC-III) has been used in European (Kioutsioukis et al., 2016) and regional level studies (Belis et al., 2020), as well as on metropolitan region (Kuik et al., 2016) and city-scale (Ramacher et al., 2021) modelling. Therefore, the GAMS-REG-inventory is a good comparison dataset for more detailed and regionally specific inventories.

A larger scale inventory can also be comprised of several independent inventories. Li et al. (2017) describes a mosaic approach, in which several national and/or regional-level inventories are combined to one. The source sectors, pollutants and spatial and temporal resolutions are normalized to create a harmonized dataset. The strength of such an inventory is the possibility to use state-of-the-art available data for each region included in the dataset. HTAP_v3 mosaic (Crippa et al., 2022) is an example of such an inventory, combining national and independent inventories to create a global emission dataset.

In this paper we present a new air pollution emission inventory for the Nordic countries with high-resolution spatial allocation (1 km $\times$ 1 km). The country total emissions are based on official national inventories, and the spatial allocation was done separately for each country, and combined into a single, mosaic inventory. The final inventory is compared to a European scale



emission inventory to study how national assessment affects the spatial distribution of the emissions. The comparison is done for two major area source sectors with low emission height, i.e. road transport and machinery and off-road mobile sources.

The third major sector, residential wood combustion, was analysed in a separate paper in Paunu et al. (2021). The research questions of this paper were:

(1) How does European level inventory describe emissions in the Nordic countries for road transport and machinery and off-road sectors compared to our more detailed national level inventory?
(2) What data is needed for high-quality spatial distribution of air pollution emissions of road transport and machinery

and off-road, and how does the EMEP/EEA guidebook cover them?

## 2. Methods

Our common Nordic (Denmark, Finland, Iceland, Norway, and Sweden) inventory (Paunu et al., 2023) was compiled using country total emissions from national emission inventories that the countries submit to the CLRTAP. Our inventory was based on the 2016-2018 submissions. Two modifications were made: (1) road transport non-exhaust PM emissions were adjusted to

better conform with Nordic traffic dust assessments (Denby et al., 2013; Norman et al., 2016); and (2) for OC emission, that are not included in the inventories, rough estimates were calculated based on expert estimates on OC/PM2.5-ratios on main SNAP level. The inventory contains annual emissions for 1990, 1995, 2000, 2005, 2010, 2012 and 2014. Components included in the inventory are: particulate matter ($PM_{10}$ and $PM_{2.5}$), black carbon (BC), organic carbon (OC), sulphur oxides ($SO_x$), nitrogen oxides ($NO_x$), carbon monoxide (CO), non-methane volatile organic compounds (NMVOC) and ammonia ($NH_3$). The

gridding was done separately for each country, using national data and gridding methods (details are given below). The spatial proxies used in the two studied sectors, road transport and machinery and off-road mobile sources, are presented in Table 1 and Table 2, respectively. Spatial proxies for other SNAP sectors are presented in supplement Table S1. The emissions were harmonized to the same sector nomenclature, i.e. SNAP, and to the EEA reference grid. Spatial resolution for the inventory is 1 km × 1 km in the European grid ETRS89-LAEA (EPSG: 3035). Large point source emissions are provided with locations

and stack heights included. The inventory was originally created for the NordicWelfAir-project (Frohn et al., 2022; Lehtomäki et al., 2020), which determined the years and pollutants included. The main aim of developing this new inventory was to provide air pollution modelers and health scientists a harmonized dataset to be used for studies on the link between air pollution exposure and negative impacts on the human health.



**Table 1 Spatial proxies used for road transport emissions in the Nordic countries.**

| SNAP code | Sector | Denmark | Finland | Iceland | Norway | Sweden | CAMS-REGv4.2 |
|---|---|---|---|---|---|---|---|
| 0701-0705 | Road transport: exhaust | Road network Traffic volume | Municipal fuel use Road network Traffic volume | Road network Traffic volume | Regional CO2 emissions Road network Traffic volume | Road network Traffic volume Statistical areas | Road network Traffic volume |
| 0706-0708 | Road transport: dust | Road network Traffic volume | Municipal vehicle-kilometres Road network Traffic volume | Road network Traffic volume | Regional CO2 emissions Road network Traffic volume | Road network Traffic volume Statistical areas | Road network Traffic volume |






**Table 2 Spatial proxies used for machinery and off-road emissions in the Nordic countries.**

| SNAP code | Sector | Denmark | Finland | Norway | Sweden | CAMS-REGv4.2 |
|---|---|---|---|---|---|---|
| 0801 | Military | Military training areas | Population | n.a. | Military exercise areas, air force & naval bases | Population |
| 0802 | Railways | Railways | Population | n.a. | Railways where diesel trains are used (70 %), marshalling yards (30 %) | Railways |
| 080501-080502 | Air traffic, LTO* cycles | Airports Aviation data | Airports | n.a. | Airports weighted by LTO statistics | Airport distribution |
| 0806 | Agriculture | Agricultural fields | Agricultural fields | Land use, agriculture, forestry | Agricultural fields, tractors per municipality | Arable land |
| 0807 | Forestry | Forests | Forest areas excl. national parks | Land use, agriculture, forestry | Tree-cuttings | Arable land |
| 0808 | Industry (including construction) | Industrial areas, Construction statistics (roads and buildings) | Industrial areas | Land use, construction site and industry | Industrial areas, iron & steel facilities, mines, population, building construction permits per county | Industrial area |
| 0809 | Household and gardening | Residential detached housing area | Residential detached housing area | Land use, urban discontinuous area | Residential detached housing area | Population |
| 0810 | Other off-road | Technical, sport, recreational areas, scrubs, cemeteries | Population Traffic volume Mining areas | Land use, urban areas | Multi-dwelling housing area Detached housing area Detached housing living space & number of vehicles per municipality Pier inventory & inland water bodies Traffic volume | Population |

*landing and take-off






**Table 3 PM$_{2.5}$ emissions and share of national sectoral total of machinery and off-road subsectors in the Nordic inventory.**

| SNAP sector | Denmark Emissions [ton/year] | Share | Finland Emissions [ton/year] | Share | Norway Emissions [ton/year] | Share | Sweden Emissions [ton/year] | Share |
|---|---|---|---|---|---|---|---|---|
| 0801 Military | 8 | 0.9 % | 15 | 1.2 % | | | 16 | 0.9 % |
| 0802 Railways | 61 | 6.4 % | 33 | 2.7 % | | | 62 | 3.6 % |
| 080501-080502 Air traffic, LTO* cycles | 11 | 1.2 % | 0 | 0.0 % | | | 29 | 1.7 % |
| 0806-807 Agriculture & Forestry | 503 | 52.6 % | 307 | 25.2 % | 211 | 31.0 % | 362 | 21.0 % |
| 0808 Industry | 341 | 35.7 % | 468 | 38.5 % | 87 | 12.7 % | 758 | 43.9 % |
| 0809 Household and gardening | 11 | 1.2 % | 12 | 1.0 % | 351 | 51.5 % | 194 | 11.2 % |
| 0810 Other off-road | 20 | 2.1 % | 380 | 31.3 % | 33 | 4.9 % | 307 | 17.8 % |
| Total | 956 | | 1215 | | 682 | | 1726 | |

\* LTO=Landing and takeoff

## 2.1. National emissions

### 2.1.1. Denmark

The spatial distribution of the emission inventory for Denmark was based on the Danish model system SPREAD (Plejdrup et

al., 2021). The SPREAD model uses an orthogonal grid with a spatial resolution of 1 km × 1 km covering the Danish area

defined by the national border on land and the exclusive economic zone (EEZ) on sea. The model includes spatial distribution

keys (GeoKeys) for emissions sources on the most disaggregated sectoral level possible. The GeoKeys are based on plant

specific data when available, and otherwise on proxy data.

The main data sources for the GeoKeys are the Building and dwelling register (BBR), the Central Husbandry Register (CHR),

the fertiliser accounts, the General Agricultural Register (GLR), data from the Danish Association of Chimneysweepers (SFL),

the national topographic object oriented map KORT10, the GIS-based National Road and Traffic Database (NRTD), and

national statistics including a spatial element, e.g. municipality. All GeoKeys are described in Plejdrup et al. (2021).

The distribution of emissions from small combustion plants is based on detailed data from the SFL, including number and type

of installation on address level. For residential wood combustion (RWC), this has been combined with information from local

studies to apply weighting factors for appliance types (stove and boiler) and building type (permanent residence, apartment

and holiday house). The methodology is described in Plejdrup et al. (2016).

The NRTD include mileage data per road segment for the Danish road network collected from a large number of local data

(Jensen et al., 2019). The model includes five road types and four vehicle types, which are aggregated to three road types and

three vehicle types in SPREAD to match the categorisation in the emission reporting. Data are included for every fifth year

180    1960-2020.

Emissions from machinery and off-road are comprise of a number of sources with separate spatial distributions and proxy data.

Emissions from military machinery are allocated to training areas for land-based machinery and to the Danish territory for

military aviation. Railway lines are used as proxy for rail transport and a 5 km buffer zone around the major airports are used

as proxy for LTO. The distribution for machinery in agriculture and forestry are based on the Danish land use matrix. Land

use data are used as proxy for commercial/institutional, residential, and industrial machinery based on KORT10. For residential

machinery the land use category "residential detached housing area" is used.

The distribution for the agricultural sector is based on detailed data from different agricultural registers. The number of animals

by type per farm from the CHR are used for GeoKeys for manure management. The GLR holds information about the fields



that are applied for subsidies, the location and size of the fields, and the crops grown on the specific field. These data in combination with information of use of fertilizers (animal manure, inorganic fertilizer and other N containing materials, e.g., sludge) from the fertilizer reports to prepare GeoKeys for agricultural soils.

All emissions included as large point sources in the Danish emission inventory, and point sources where plant specific data are available, are treated as point sources in the SPREAD model, and the emissions are allocated to the exact location. Among the data sources for point sources are EU-ETS reports, annual accounts for individual companies and plants, and the annual database from the Danish Energy Agency (DEA), holding fuel consumption and technology information separately for each district heating or power production plant.

### 2.1.2. Finland

Spatial distribution of the emission of Finland was based on the Finnish Regional Emission Scenario (FRES) model (Karvosenoja, 2008). Spatial resolution of the FRES model is 250 m × 250 m, from which the emissions were aggregated to the 1 km × 1 km resolution for the Nordic inventory.

Main data sources for spatial proxies for the Finnish emissions are CORINE Land Cover (CLC) 2012 data (European Union, 2012), Finnish Building and Dwelling Register (BDR) and national road and street database Digiroad. Spatial distribution of residential wood combustion is presented in Paunu et al. (2021). For road transport (Table 1), national emissions are first distributed to municipalities with municipal fuel use data (for exhaust emissions) and vehicle-kilometre data (for non-exhaust, i.e., traffic dust). Distribution is done separately for trucks, buses, vans, and passenger cars using vehicle type specific data. From municipal emissions to grid, road locations from Digiroad are combined with measured traffic volume from Digiroad (for highways) or relative street-class specific traffic volumes based on municipal traffic counts (for streets). For agriculture, emissions from animals are distributed to municipalities according to animal numbers, and to the grid according to field area. Other agriculture emissions are distributed with field area. Emissions from peat production are distributed according to peat production area.

Emissions from the machinery sector are a combination from several subsectors (Karvosenoja et al., 2020). For spatial distribution, the emissions were compiled into 11 subsectors. SNAP sectors 0808 and 0810 for industrial and other off-road, respectively, were regrouped into industrial machinery, other off-road, street maintenance machinery, road maintenance machinery, and mining machinery. Proxies used are presented in Table 2. Most subsectors were distributed with land use data using the area of the land use class as the proxy. Street and road machinery used roads and traffic volumes, the same data as the road transport sector. Few subsectors were distributed with population data, as other suitable proxy data was not identified at the time.

Point source emissions for Finland were calculated using the FRES model. The calculation was based on the average annual fuel use or production of the power and industrial plants. The annual value was defined individually for each year, based on three-year average around the year. The total sectoral emissions were compared to the official national inventory, and remaining emissions were divided into municipalities according to fuel use of small municipal energy production facilities and treated as point sources in the centre of the municipality.

### 2.1.3. Iceland

The emission inventory for Iceland covers road transport (exhaust and road wear) and point-source emissions. Road transport represents the main air pollution source for the capital area, and point sources such as aluminum smelters and geothermal power plants for the whole country. In contrast to other Nordic countries, residential wood combustion is rare, and mainly occurs to create cozy atmosphere ("hygge") in some recreational houses. Data on machinery and construction sites were lacking and were thus not included.





Road transport exhaust emissions were based on the national inventory, except for PM emissions, which were based on

Emissions Database for Global Atmospheric Research (EDGAR) (Crippa et al., 2018) for year 2008. Non-exhaust emissions were calculated by weighting results from SIMAIR for Sweden. The spatial distribution of the road transport emissions was based on traffic modeling (Hreggviðsson, 2012).

### 2.1.4. Norway

relevant sectors for air pollution and population exposure. The selected sectors were combustion in energy and transformation

industries (SNAP1), combustion in manufacturing industry (SNAP3), production processes (SNAP4), residential heating (SNAP2) focusing on wood burning emissions, solvent and other product use (SNAP6), road transport (SNAP7), agriculture (SNAP10), and other mobile sources and machinery (SNAP8), focusing on mobile combustion machinery in manufacturing industries and construction, commercial/institutional, household and gardening and in agriculture/forestry/fishing.

Emissions from SNAP1, SNAP3 and SNAP4 were obtained from the Norwegian Pollutant Release and Transfer Register (NO

PRTR) and spatially distributed based on the exact location of the large point sources provided by the Norwegian Environment Agency. Emissions from residential heating (SNAP2) is largely dominated by wood burning in small-scale combustion for residential heating. Emissions were estimated based on wood consumption data per type of technology (i.e., open fireplace, closed wood stove produced before 1998, and closed wood stove produced after 1998) at county level from Statistics Norway for each year, emission factors developed for Norwegian conditions (Seljeskog et al., 2013) and dwelling number at 250-meter

resolution for each emission year (for more detail see Paunu et al., (2021)).

National emissions from the on-road transport (SNAP7) were spatially distributed on the road network using data from the Norwegian Road Administration and gridded at 1 km × 1 km. The spatial distribution was based on 1) $CO_2$ emissions at regional level; 2) the pollutant of interest to $CO_2$ ratio at national level for heavy (HDV) and light duty vehicles (LDV); and 3) the amount of traffic at the road links expressed as average daily traffic along with the share of HDV. The $CO_2$ emission

data at regional level was used as proxy to account for the heterogeneous distribution of technological vehicle class in the Norwegian geography, and thereafter the emissions were distributed on the road network based on the average daily traffic. Based on the ratio $CO_2$ to pollutant of interest determined at national level for HDV and LDV weighted by the kilometres driven by passenger car and other LDV, we spatially distribute the emissions. In the case of non-exhaust emissions, the same procedure was followed.

Emissions from agriculture (SNAP10) were distributed using land use data classified as agriculture in CORINE Land Cover 2006 (European Union, 2006). Emissions from other mobile sources and machinery (SNAP8) officially reported under the obligations of CLRTAP were distributed based on land use classifications; construction and industry land use data from CORINE classification, urban areas in the case of emissions from commercial/institutional mobile sources, discontinuous urban areas in the case of household and gardening, and agriculture forestry in the case of mobile machinery in

agriculture/forestry/finishing.

### 2.1.5. Sweden

Details of the spatial allocation of Swedish emissions are described in Andersson et al. (2019). For small scale residential heating the methodology is also described in Paunu et al. (2021). In general, spatial allocation is made at a low level of aggregation allowing spatial proxies to be as specific as possible. The spatial resolution or the spatial proxies is 1 km × 1 km

and the spatial reference system is SWEREF 99 TM. For use in the Nordic inventory reprojection was required, resulting in a slightly reduced spatial precision (an increased spatial uncertainty in the range 0-500 m). In total around 60 spatial proxies are combined with statistics at the lowest applicable level of aggregation generating around 100 unique spatial distributions for around 190 emission sectors. In short, road transport emissions have been distributed based on a national bottom-up emission calculation. Traffic flow is based on measurements and a national traffic model provided by Swedish Road Administration.

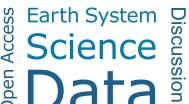

The composition of the vehicle fleet on each road is based on national statistics from the Swedish Transport Administration and is assigned to each road based on road type, if it's an urban or rural area, and using share of heavy vehicles as described by the national traffic model. Emission factors from HBEFA 3.2 have been used. A minor share of the emissions has not been allocated to individual roads but is instead described for statistical areas employed by the Swedish Road Administration. The bottom-up estimates are updated less frequently than national total emissions. To ensure consistency, the bottom-up estimates

for different substances have been gridded at 1 km × 1 km resolution and are used as spatial proxies.

A refined split has been made of total emissions for different types of machinery, allowing separate proxies to be used for each type. For example, for construction machinery, area of building permits per year are used to distribute emissions between municipalities, and population density within each municipality. For mining industry, emissions are allocated to mines based on statistics on total amount of extracted ore. For forestry machinery, a yearly gridded description of volume of logged wood

based on a remote sensing analysis provided by the Swedish Forest Administration is used to allocate the emissions. For agricultural machinery, tractors are distributed between municipalities based on statistics and then allocated to farmland. Machinery used at harbours and airports are distributed between individual locations based on statistics of activity (arrivals or departures). Emissions from road construction machinery is distributed using gridded road work as a proxy.

Trend analysis is carried out for each sector on a regional level to ensure consistency over time in the spatial distribution of

emissions. Due to statistical confidentiality, the majority of the point-sources is not represented as individual sources, but instead included in the gridded emissions. For the point sources that are provided individually, the stack height is estimated based on the size of the emissions.

### 2.2. Shipping emissions

Shipping emissions included in the Nordic inventory were calculated using STEAM v2 (Jalkanen et al., 2012; Johansson et

al., 2013). STEAM describes shipping activity/distribution using positioning data from AIS (Automatic Identification System). Use of AIS increased gradually since it's development in the 1990's. Since 2002, AIS is required for ships over 300GT according to IMO (the International Maritime Organisation) regulations. The emissions from ships are based on the European 2011 AIS data described in Jalkanen et al (2016). The 2011 data was used as a starting point (Table 4), and annual emissions were scaled to reflect one percent growth in $CO_2$ emissions each year when applied for past years (decreasing future trend run

backwards). When fuel sulphur regulations changed in 2010 (EU Sulphur Directive (EU) 2016/802, IMO MARPOL Annex VI) and $SO_x$ Emission Control Areas (North Sea (2007), Baltic Sea (2006)) became effective, these impacted $SO_x$ and PM emissions. In the calculations, the fuel type assignment (residuals, distillates) was kept constant, and fuel sulphur content was changed to mimic the effect of sulphur emission control areas (SECA) (Table 5). Fuel-based emission factors were from 3rd IMO GHG study, Annex 6, Table 22 (IMO, 2014) (assuming an average specific fuel oil consumption of 200 g/kWh). The

fuel-mix is assumed constant over the whole time-period, with 75 % HFO (Heavy Fuel Oil), 24 % MDO (Marine Diesel Oil) and 1 % MGO (Marine Gas Oil). Before year 2000, no changes in average sulphur content were assumed. It should be noted that using 2011 ship activity data for Europe and scaling emission totals without considering dedicated sea area specific calculations may lead to uncertainties in the emissions. Also, projections to past, as distant as 30 years, is also prone to significant uncertainty. Our approach aimed to describe the SECA zones more accurately than the non-SECAs.

Changes in $NO_X$ emission factors due to fleet renewal were included by a linear interpolation of $NO_X$ emission factors of old engines to emission factors of new engines. There were large uncertainties in VOC emissions. Assuming gradual technology improvements, estimates in the upper end of the range found in literature have been applied for older ship engines (0.0023 g/g (Entec UK Limited, 2002) and estimates from the lower end for newer engines (0.00045 g/g (Cooper, 2001)). Annual growth is assumed 1 % per year before year 2000.




**Table 4 CO₂-content of different fuel grades applied in estimations for the years 1990-2000 using year 2011 as baseline.**

|          | Heavy fuel oil | Marine diesel oil | Marine gas oil |
|----------|----------------|-------------------|----------------|
| $CO_2$, g/g | 3.1147 | 3.206 | 3.206 |

**Table 5 Fuel sulphur content applied in estimations of emissions for years 1990-2000 using year 2011 as baseline.**

|                   | 1980  | 1986  | 1991  | 1996  | 2001  | 2006  | 2011  |
|-------------------|-------|-------|-------|-------|-------|-------|-------|
| Heavy fuel oil    | 2.4 % | 2.4 % | 2.4 % | 2.4 % | 2.4 % | 1.5 % | 1.0 % |
| Marine diesel oil | 0.5 % | 0.5 % | 0.5 % | 0.5 % | 0.5 % | 0.5 % | 0.5 % |
| Marine gas oil    | 0.1 % | 0.1 % | 0.1 % | 0.1 % | 0.1 % | 0.1 % | 0.1 % |

### 2.3. CAMS-REG v4.2 inventory

CAMS-REGv4.2 is a European scale emission inventory to support air quality modelling exercises, and version 4.2 covers the years 2000-2017. This inventory is described in detail in Kuenen at el. (2022). To support the policy applications, the inventory
relies where possible on the official reported emission data to the Convention on Long-Range Transboundary Air Pollution (LRTAP) as well as the EU National Emission Ceilings Directive (NECD), based on the 2019 submissions. These official reported data are collected from the national reporting of air pollutants and checked for inconsistencies, gaps, and errors. In those cases where for a specific country and pollutant the reported data were found unfit for use, these were replaced with other emission estimates, most notably from the IIASA GAINS model (Amann et al., 2011). The spatial resolution of the
inventory is 0.1° × 0.05° (longitude-latitude, roughly 5.5 km × 3.5-6.5 km in the Nordic countries).

The CAMS-REGv4.2 inventory uses the country reported data of the sectoral emissions at national level, while the spatial distribution is performed using an independent and consistent approach over Europe. This implies that for each sector, a relevant distribution parameter is chosen. For road transport the spatial distribution of emissions is based on a combination of Open Street Map for the road network, and Open Transport Map providing information on the traffic volumes. The spatial
distribution parameters for each detailed sector in the inventory is described in Kuenen et al. (2022) and the supplementary information therein. Shipping emissions at sea in CAMS-REGv4.2 are based on the STEAM model approach (described in Section 2.2). CAMS-REGv4.2 uses a more recent version of STEAM (v3.5) than the Nordic inventory introduced in this paper, with more complete AIS datasets and more detailed consideration of regional sulphur rules. For other non-road transport, railway emissions are allocated to a map of the rail lines, whereas airports are assigned to the location of main airports weighted
by the number of flights per airport. This is done for each individual year to reflect opening and closure of specific airport sites. For mobile machinery used in industry, emissions are assigned to industrial areas from Corine Land Cover dataset (CLC2012) (European Union, 2012), whereas for agricultural the arable land classification from the same dataset is used to spatially disaggregate emissions. Remaining "other" sources are spatially distributed using population density.

### 2.4. Comparison methods

A statistical comparison between the Nordic and CAMS-REGv4.2 inventories was carried out using the index of agreement developed by Duveiller et al. (2016). The index is non-dimensional, bounded and symmetric. The index ($\lambda$) is calculated as

$$\lambda = \alpha \cdot r = \frac{2}{\sigma_X/\sigma_Y + \sigma_Y/\sigma_X + (\bar{X}-\bar{Y})^2/\sigma_X\sigma_Y} \cdot r, \tag{1}$$

where $\bar{X}$ and $\bar{Y}$ are the mean values of the datasets $X$ and $Y$, $\sigma_X$ and $\sigma_Y$ are their standard deviations, and $r$ is the Pearson product-moment correlation coefficient. $\alpha$ represents how much the index of agreement deviates from the Pearson correlation



coefficient. The index also enables the separation of unsystematic and systematic differences, represented by unsystematic index $\lambda_u$ and systematic noise $f_{sys}$. The unsystematic index $\lambda_u$ represents the value the index $\lambda$ would take if there was no systematic bias between the datasets (for example one dataset having systematically higher values than the other). The systematic noise $f_{sys}$ represents the proportion of total deviation caused by the systematic bias. The details of the method and the mathematics can be found in the reference. The index of agreement was calculated with those cells that had non-zero values

in either dataset. This reduced the number of no data cells and highlighted the differences between the datasets.

The spatial comparison was carried for two sectors, road transport and machinery and off-road. Road transport sector is a good example of sector in which the spatial distribution methods are well established, and data is readily available (Table 1). In contrast, machinery and off-road is more difficult due to its heterogeneity and less obvious suitable proxies (Table 2). For the comparison the Nordic inventory was aggregated to the same resolution as the CAMS-REGv4.2 inventory ($0.1° \times 0.05°$). The

comparison was done separately for each country. Furthermore, a visual comparison was done to complement the analysis. The $PM_{2.5}$ emissions were compared to assess their role in the analysis of the differences in the spatial distribution. Finally, differences in normalized proxy weight maps were visually analysed. The emissions were normalized separately within each country by dividing the cell values by the total country emission of the sector. The difference maps were then calculated for each country raster.

## 3. Results


In the following we first present the total emissions and the spatial distributions of road transport and machinery and off-road emissions in the Nordic inventory, and thereafter the comparison to the European dataset.

Total Nordic emissions for all pollutants are presented in Figure 1 (shipping not included). Shipping emissions are presented in Figure 2. Figure 3 presents the spatial distribution of the total $PM_{2.5}$ emissions for 2014, and also shows the spatial coverage

of the shipping emissions in the inventory.

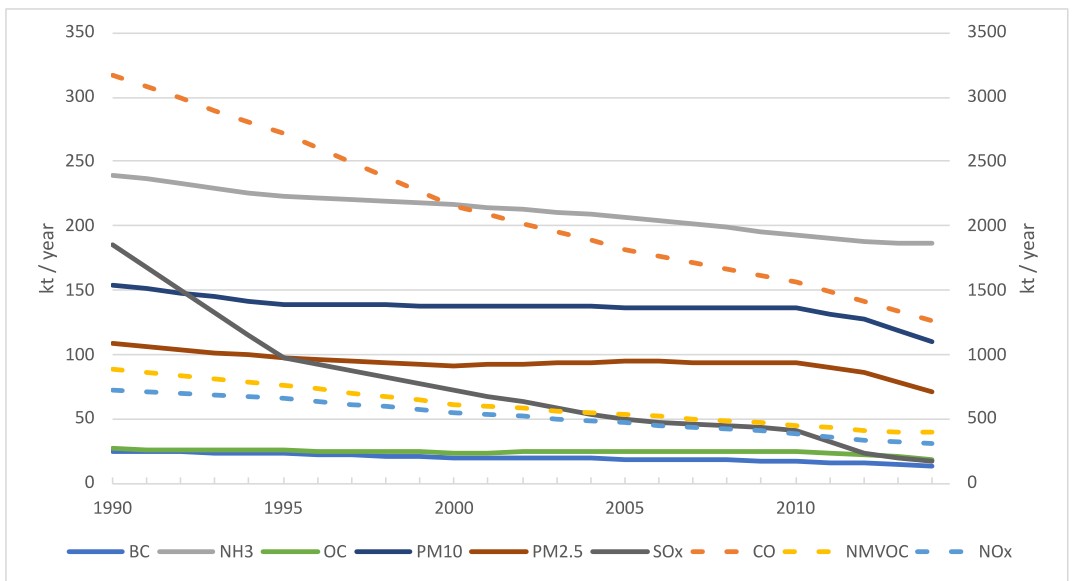

**Figure 1 Total Nordic emissions for 1990-2014 (shipping excluded). CO, NMVOC and NOx emissions (dashed lines) use the right y-axis (which is 10 times the left axis).**


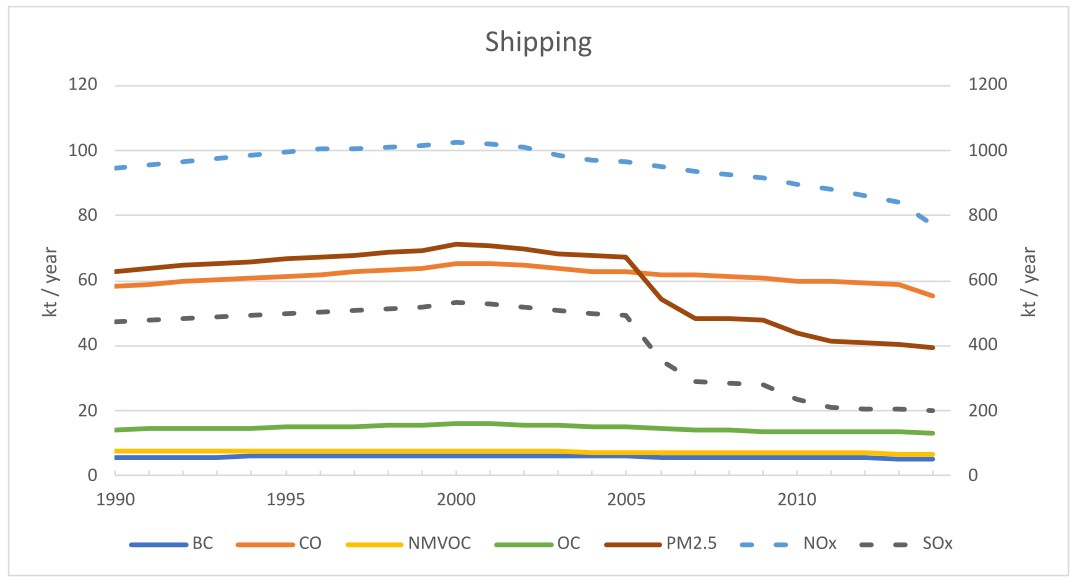

**Figure 2 Shipping emissions in the Nordic inventory for 1990-2014. NOx and SOx emissions (dashed lines) use the right y-axis (which is 10 times the left axis). For spatial extent of the shipping emissions, see Figure 3.**

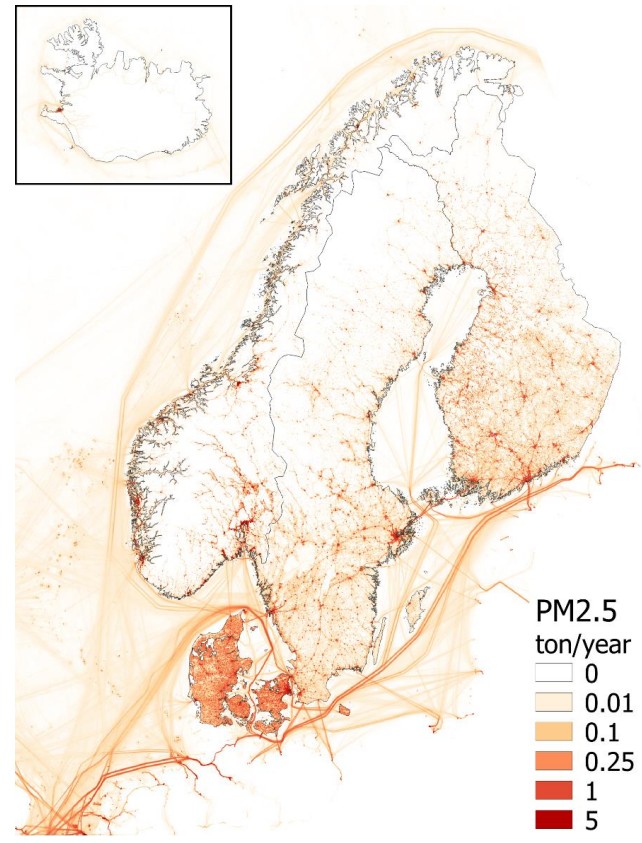


**Figure 3 Total PM$_{2.5}$ emissions in the Nordic countries in 2014 (except shipping around Iceland in 2010) in 1 km x 1 km resolution. The map also shows the spatial extent of shipping emissions in the Nordic inventory.**



### 3.1. Spatial distribution of road transport emissions

Spatial distribution of PM$_{2.5}$ emissions from road transport is presented in Figure 4. The methods and spatial distributions were
well aligned between the countries. All countries used road network data and traffic volumes as proxy data. Each country also took distribution between heavy and light duty vehicles into account. Sweden and Norway had two vehicle categories, Denmark used three, and Finland separated the vehicles into four categories (two heavy and two light duty). The countries had separate proxy data for each category. The traffic volume data were measured where possible and modelled to fill the gaps. Often the roads were divided into several types, which then were assigned with relative traffic volumes. Finland and Norway
used a two-step process, in which the emissions were first distributed to regional or municipal level, and then to the road network within those areas. Finland took the vehicle categories account on municipal level, whereas Norway did it on national and again on road level. Sweden allocated minor part of the emissions to larger statistical areas instead of roads, creating large areas with uniform, small emissions. In each country the emissions were concentrated to larger cities and busiest highways, as can be expected. Denmark had the most evenly distributed emissions due to higher population density, and, thus, higher density
of roads, than the other Nordic countries. Spatial distributions for road transport exhaust and dust emissions were similar in each country. Finland had different municipal-level proxies for exhaust and dust emissions, former being based on fuel use and latter on kilometres driven. In practice, the two distributions were very similar.

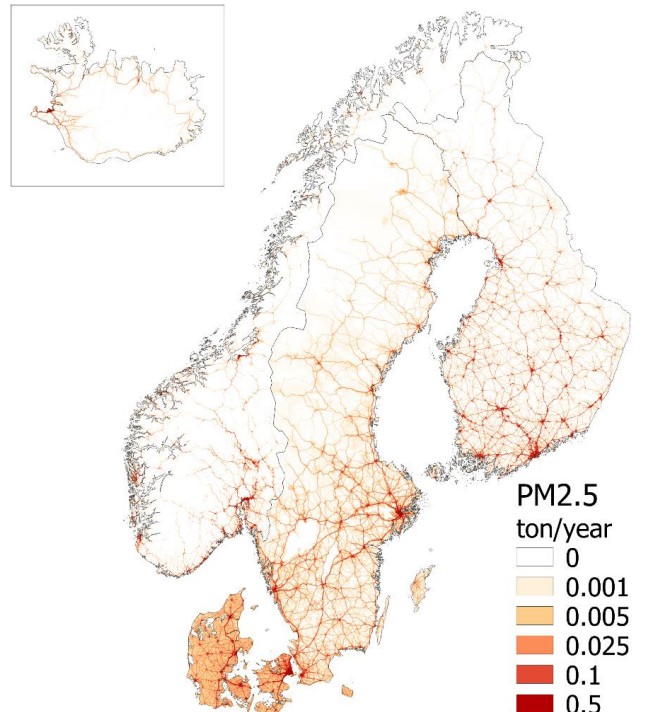

**Figure 4 Road transport (SNAP 7) PM$_{2.5}$ emissions in the Nordic countries in 2014 in 1 km x 1 km resolution.**

### 3.2. Spatial distribution of machinery and off-road emissions

**Spatial distribution of PM$_{2.5}$ emissions from machinery and off-road is presented in Figure 5.**



Table 3 shows the sub-sectoral emissions and their share of the national total of the sector. Spatial distributions for machinery emissions showed more intricate differences between the countries than road transport emissions. One reason for this was that the machinery and off-road sector that was considered here consists of several heterogenous subsectors, and, therefore, contains multitude of proxies. The proxies used are presented in Table 2. The emission shares of the subsectors were variable between the countries. In 2014, in Denmark the highest shares of machinery emissions were agricultural & forestry (53%) and industry

(36%). In Finland the highest shares were industry (39%), other off-road (31%) and agriculture and forestry (25%). In Sweden the order was industry (44%), agriculture and forestry (21%) and other off-road (18%). Norway was the only one with highest share in household and gardening (52%), followed by agriculture and forestry (31%). However, there were some differences in where construction and road maintenance emissions are allocated regarding the subsectors (presented in the Table 2). Denmark allocated all of them under industry, but Sweden and Finland included road construction and maintenance under

other off-road.  While this is mainly a technical detail, it highlights the complexity of the machinery and off-road sector. Even with some differences in the allocation, the shares indicate how the countries differed from each other regarding the estimation of the underlying activities and their emissions.

    In all countries the most emissions were weighted to cities. In addition, in Finland roads stood out, and a few mines had high emissions. In Sweden, notable areas were large railway stations and their surroundings, airports, and some mines. In Norway,

there were little emissions outside the population centres. Denmark had the most even distribution. Some larger roads, especially in the east, had high emissions on highways due to road construction.

    Land use was the most common proxy data used in machinery and off-road. In some cases, statistical data were used to weight the emissions with more than just the area of the land use class. For example, Sweden combined registered tractors per municipality with agricultural field area, and Denmark used construction statistics of roads and buildings. One of the most

prominent differences was how roads were used for proxies. Norway didn't use road data for machinery, and Finland had most weight on them of the four countries, apart from a few road segments in Denmark. It is notable that population was used very little as proxy. Only Finland distributed some maintenance and construction machinery that way.

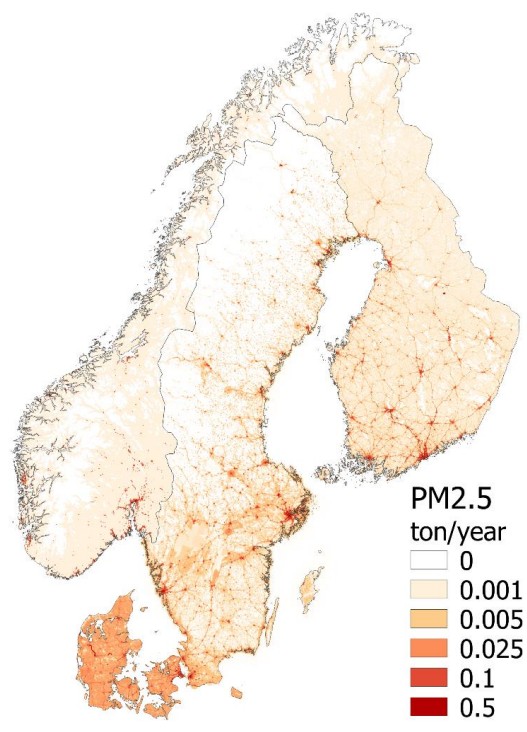

**425**     **Figure 5 Machinery (SNAP 8) PM$_{2.5}$ emissions in the Nordic countries in 2014 in 1 km x 1 km resolution.**

### 3.3. Comparison to CAMS-REGv4.2

#### 3.3.1. Country total emissions

The total national PM$_{2.5}$ emissions from road transport and machinery and off-road sectors for 2010 in the Nordic and CAMS-REGv4.2 inventories are presented in Table 6. For road transport, the emissions were similar, except for Iceland and Sweden.

**430** In Iceland, the significant difference was related to the use of EDGAR emissions for PM emissions from road transport in the Nordic inventory, whereas the CAMS-REG inventory used the official national inventory results from Iceland. For Sweden, the difference came from traffic dust emissions. These emissions were adjusted in the Nordic inventory from the official reported emissions (as explained in the methods section), and the CAMS-REGv4.2 follows the official reported numbers, so the difference came from estimation methods. In this paper we won't go into further details on this, as we focus on spatial

**435** distribution of the emissions.

For machinery the emissions were similar, except for Norway. In Norway, the Nordic inventory focused on the main subsectors, and, therefore, lacks emissions from some less significant subsectors. The CAMS-REGv4.2 also had higher emission estimates in the household and gardening-subsector, where the estimate was as large as the whole sector estimate for Norway in the Nordic inventory.

**440**

**Table 6 PM$_{2.5}$ emissions from road transport and machinery and off-road sectors in the Nordic and CAMS-REGv4.2 inventories for 2010.**

| PM$_{2.5}$ (ton) | | Nordic | CAMS-REGv4.2 | %-Diff |
|---|---|---|---|---|
| Road transport | Denmark | 2481 | 2481 | -0.01 % |
| | Finland | 2820 | 2811 | -0.3 % |
| | Norway | 1907 | 1878 | -2 % |



|  | | Nordic | CAMS-REGv4.2 | Difference |
|---|---|---|---|---|
|  | Sweden | 2982 | 4216 | 41 % |
|  | Iceland | 228 | 147 | -36 % |
| Machinery & off-road | Denmark | 1418 | 1523 | 7 % |
|  | Finland | 1537 | 1761 | 15 % |
|  | Norway | 758 | 1254 | 65 % |
|  | Sweden | 1705 | 1508 | -12 % |

### 3.3.2. Spatial distribution of emissions

The spatial distribution of road transport emissions in the two inventories are presented in Figure 6, and index of agreement results for the spatial distributions in Figure 7. Differences in the normalized proxy weights are presented in Figure 8.

The index of agreement $\lambda$ indicated that for road transport the distributions were similar in Finland, Iceland and Norway, whereas Denmark had lowest index of agreement. In Sweden there were two cells which had high emissions, but only in one of the inventories. Also, the cells were not close to each other, as in the Nordic inventory the high emission cell was in the

centre on Stockholm, and in CAMS-REGv4.2 it was in the centre of Malmö. Without these two cells the index of agreement results would have been nearly as good as for Finland (with $\lambda$=0.724 and $\lambda_u$=0.810). These were also the cells with largest differences in the proxy weights between the two inventories in Sweden. The Copenhagen area in Denmark was another region with differences of similar magnitude. In Iceland, the differences were even bigger in Reykjavik (Nordic inventory having higher weight), Akureyri and next to Keflavik airport (in both CAMS-REGv4.2 had higher weight).

In all countries, the Nordic inventory had more emissions weighted to the capital areas and highways, while CAMS-REGv4.2 had more weight in other cities and non-urban areas without highways. In Copenhagen this can especially be seen, as the Nordic inventory has more emissions in the city, and CAMS-REGv4.2 on the outskirts. Helsinki had few exceptions, as there CAMS-REGv4.2 inventory had much higher weight in the cell containing the city centre and also in one cell in the east (containing a large harbour), and the Nordic inventory had higher weight in the cells containing ring roads. In Finland, and to

lesser degree in Sweden, CAMS-REGv4.2 had more weight in city centres outside the capital metropolitan area. In general, in the continental Nordics the highest proxy weight differences were in cities and a few busiest highways in Finland and Denmark. Finally, there was a clear difference in the Gotland, where CAMS-REGv4.2 had higher proxy weight and emissions throughout the island.

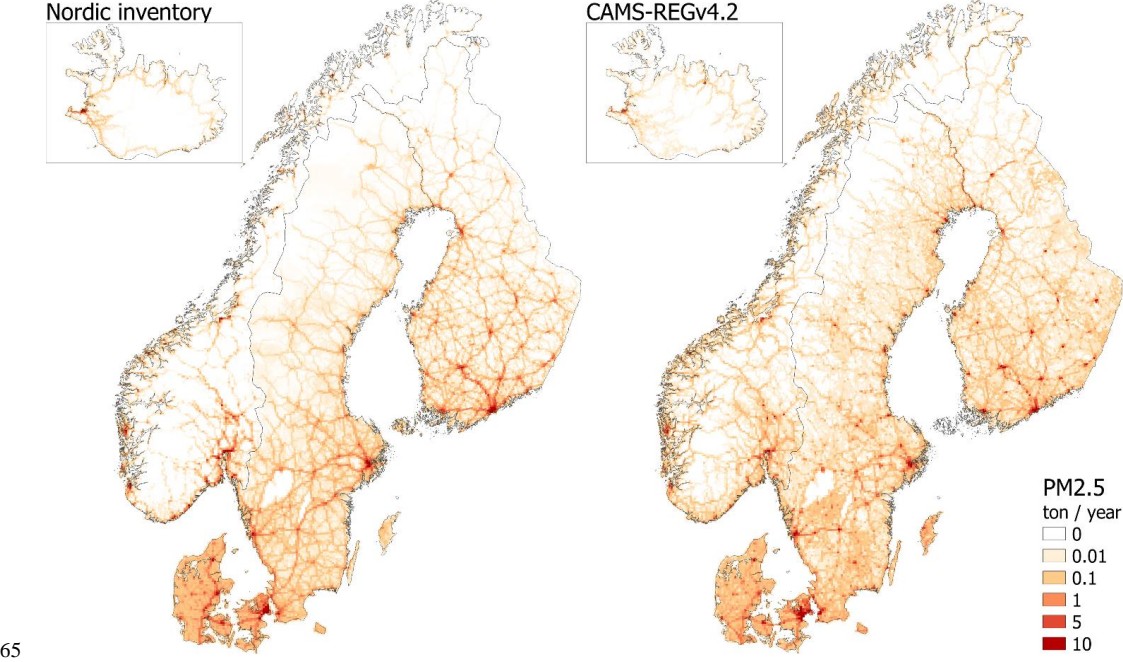


**Figure 6 Spatial distribution of road transport PM$_{2.5}$ emissions in the Nordic inventory on the left and CAMS-REGv4.2 on the right.**



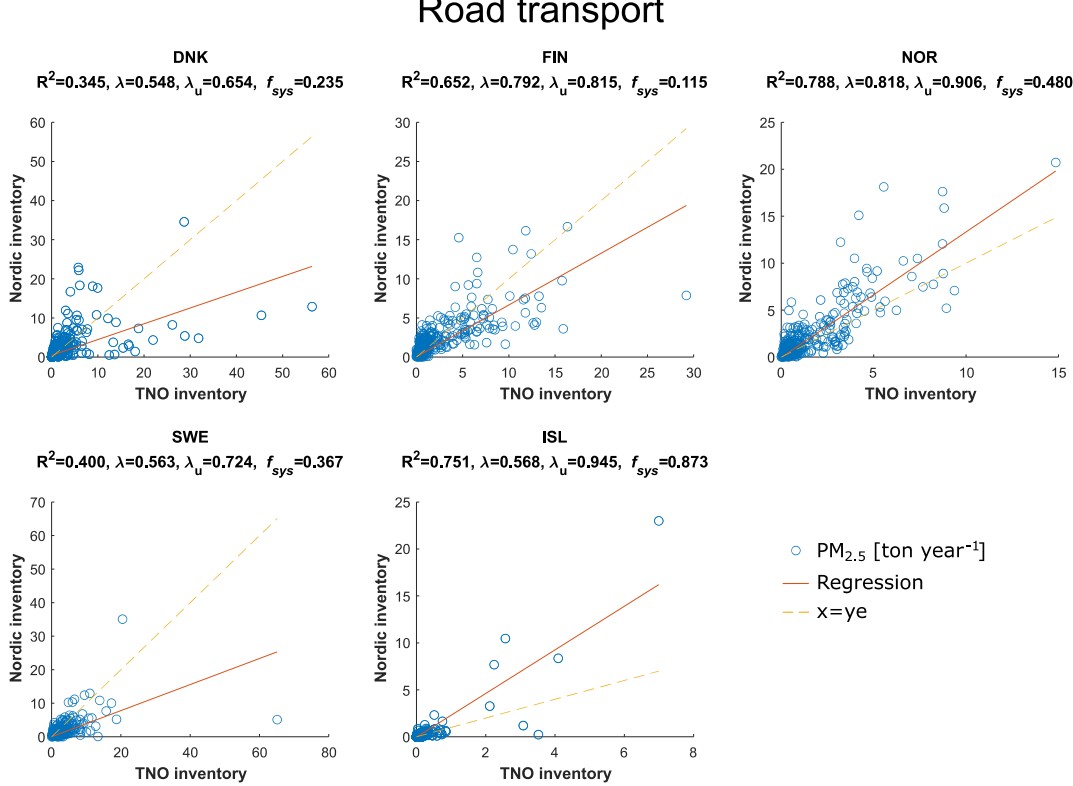

**Figure 7 Scatter plots and index of agreement results for road transport sector between our Nordic and the CAMS-REGv4.2 inventory. The circles represent emissions PM$_{2.5}$ of a single cell. Orange line is the regression and dashed yellow the y=x-line. $R^2$ = coefficient of determination, $\lambda$ = index of agreement, $\lambda_u$ = unsystematic index, and $f_{sys}$ = systematic noise.**

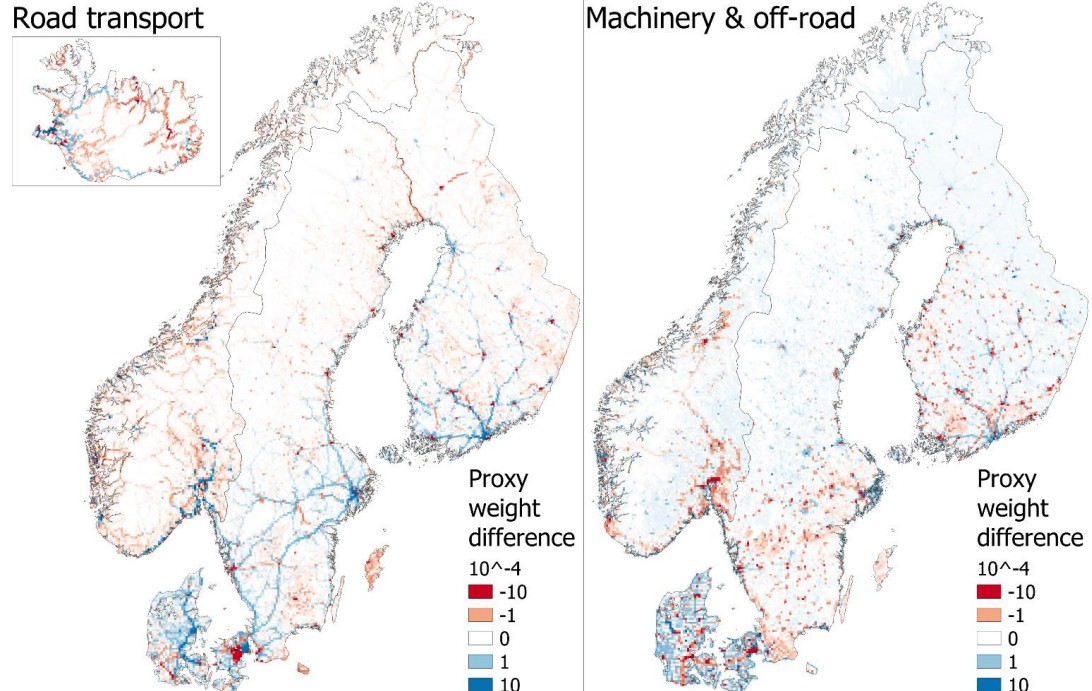

**Figure 8 Differences in the PM$_{2.5}$ emission proxy weights between the Nordic and CAMS-REGv4.2 inventories. Positive (blue) values indicate cells where the Nordic inventory had higher weight, and negative (red) same where CAMS-REGv4.2 inventory did. The weights have been normalized separately within each country.**

For machinery, the index of agreement results were poor for Denmark and Norway. For Finland and Sweden they were higher, almost as good as the best values for road transport. For Norway, the main reason was the difference in the total emissions, as seen by $f_{sys}$, which indicates the portion of systematic difference in the index. If the emissions were similar, the index would be the highest from all the Nordic countries.

In Finland, Norway, and Sweden both inventories weighted most emissions to cities. The CAMS-REGv4.2 inventory had more weight in most cities, except Helsinki and few cells in Stockholm. It also had more weight in agriculture land in all countries but Denmark. Outside city centres and agricultural fields CAMS-REGv4.2 had small emissions. The Nordic inventory had not only more emissions but also more cells with emissions (albeit low) in non-agricultural rural areas. Also, the Nordic inventory had higher weight on highways and mines in Finland, and in a few railroad stations and their surroundings and mines in Sweden. In Denmark, the Nordic inventory had more even distribution with higher weight in the rural, agricultural areas, and on a few highways (due to the construction of them). CAMS-REGv4.2 had higher weight on main highways and cities, especially in the Copenhagen area. In general, the CAMS-REGv4.2 used population as a proxy much more than the Nordic inventory.

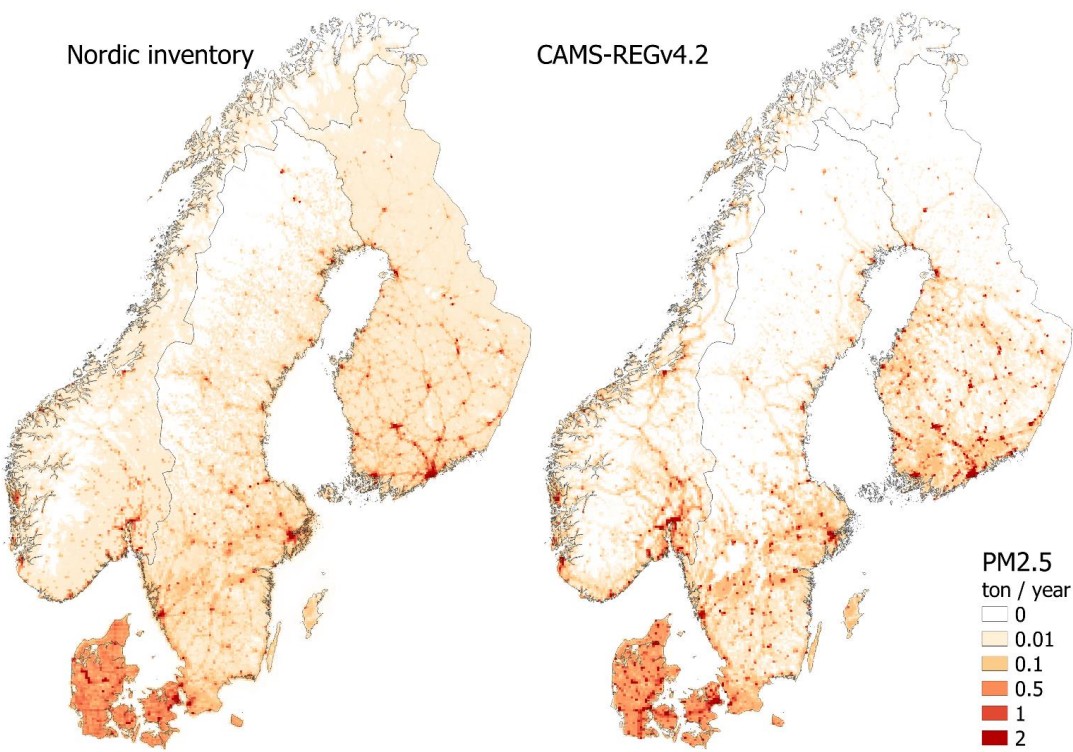

**Figure 9 Spatial distribution of machinery and off-road PM₂.₅ emissions in the Nordic inventory on the left and CAMS-REGv4.2 on the right.**




## Machinery and off-road

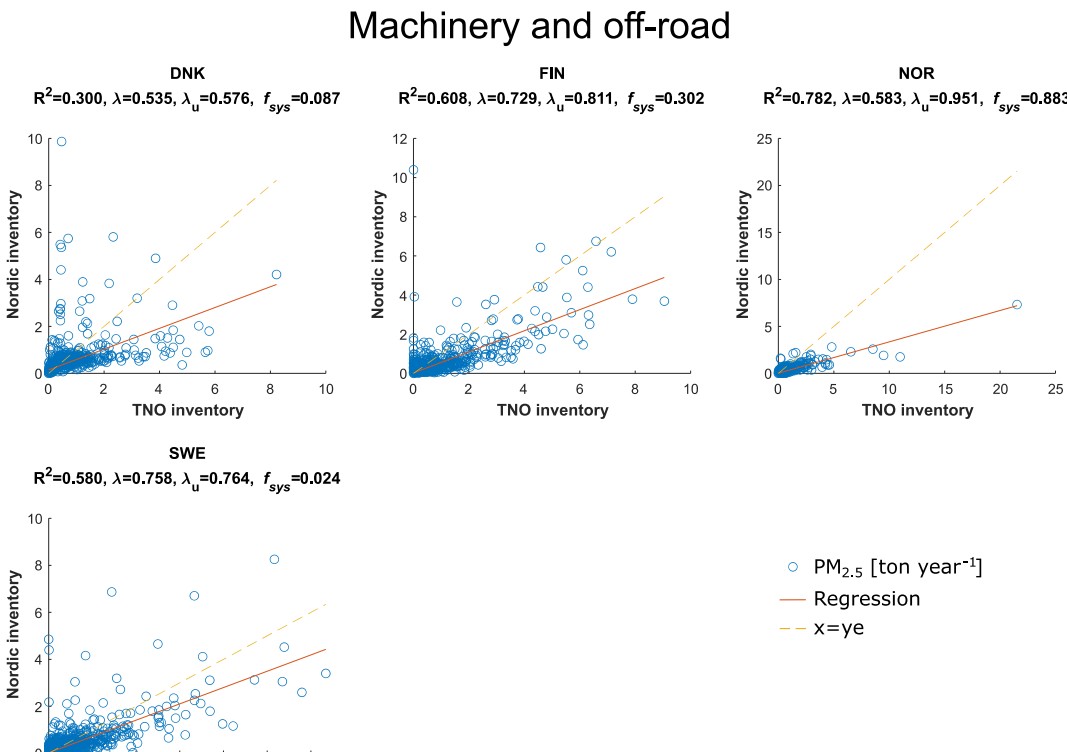

**Figure 10 Scatter plots and index of agreement results for machinery and off-road sector between our Nordic and the CAMS-REGv4.2 inventory. The circles represent PM$_{2.5}$ emissions of a single cell. Orange line is the regression and dashed yellow the y=x-line. $R^2$ = coefficient of determination, $\lambda$ = index of agreement, $\lambda_u$ = unsystematic index, and $f_{sys}$ = systematic noise.**

**4. Discussion**

Our paper describes how the Nordic countries spatially distribute the air pollution emissions of road transport and machinery and off-road sectors. In the following we will discuss the similarities and differences in methods and data that can be used to do this.

**4.1. Road transport**

The EMEP/EEA guidebook (European Environment Agency, 2019) uses the NFR nomenclature, which differs somewhat from the SNAP nomenclature used in our Nordic inventory. For road transport the guidebook recommends traffic flows and types of vehicles as the best tier spatial proxy. In each country of our Nordic inventory either measured or estimated traffic flow was used, and vehicles categories were separated at least between heavy and light duty transport. Therefore, our inventory complies with the tier 3 recommendation. The main differences between the countries were the number of vehicle categories (from two

in Norway and Sweden to four in Finland), and how much measured traffic flow data was available (and how much modelled traffic was needed to complete the proxies).

The EMEP/EEA guidebook doesn't have separate recommendations for traffic dust emissions (brake and tyre wear, road abrasion, resuspension (not required for reporting, and not included in the guidebook)). However, due to climate conditions, traffic dust is an important emission category in the Nordic countries. The emissions spike in spring, when snow melts, and

(studded) winter tyres wear the road surface and also accumulated dust is resuspended into the air. The traffic exhaust and dust



emissions were mostly distributed with the same proxies in the Nordic countries, although Finland had separate municipal level proxy for them. However, regional weather conditions, which have strong impact on the dust emissions, could be taken into account within each country. NORTRIP road dust emission model (Denby et al., 2013) contains the effect of weather to the emissions, and could be used to create regional weights for dust emissions. It should be noted that Denmark differs from

rest of Nordics, as its climate resembles Central Europe more than the other Nordic countries. This highlights that the spatial weights should be determined regionally within each country with local weather parameters.

The comparison of our Nordic inventory to the CAMS-REGv4.2 shows how combination of locally measured and modelled traffic volume data compares to a more general modelled data. Overall, the spatial distribution with the more general data was similar to the more detailed results in the Nordic inventory. It seems that highways were more weighted in the Nordic inventory,

suggesting that the traffic modelling used in the Open Transport Map, which CAMS-REG is based on, doesn't catch the traffic volumes on busy large roads well enough. Interestingly, CAMS-REG also had lower weight in capital city centres than the Nordic inventory, but higher weights in the outskirts of the capitals and smaller cities. An exception was Helsinki, where the CAMS-REGv4.2 inventory had higher weight in the cell covering the city centre. The Nordic inventory weighted more traffic and emissions on the ring roads of the Helsinki Metropolitan Area. It seems that the Open Transport Map traffic modelling

creates activity based on population density, and around transport hubs such airports and harbours. However, it does not allocate high traffic activity in the capital cities (except in the centre of Helsinki). All these could affect air pollution health impact modelling results, as densely populated areas are most important for exposure modelling results.

### 4.2. Machinery and off-road

For machinery and off-road the EMEP/EEA guidebook includes recommendations within broader sectors. For example,

industrial machinery is in the same group as stationary combustion in manufacturing industries and construction: other. The recommended proxies are often grouped in the same way, and not necessarily specific to machinery and off-road activities. It should be noted that we have used higher resolution (1 km × 1 km) than in the official gridded emission data reporting (0.1° × 0.1°) targeted by the Guidebook chapter on spatial emissions mapping. The resolution will affect what type of data can represent the activities on the chosen scale.

For industrial and commercial machinery, the guidebook recommends using point source data where possible. For industry, our inventory mainly uses industrial areas from land cover data, which is tier 1 in the guidebook. We argue that land cover is more suitable than point sources for industrial machinery, as the area where the machinery is used is better covered with such data. Point source specific production numbers, for example, could be included in the proxy to weight the emissions between the industrial areas, but to enhance the spatial distribution this would need information on industry-specific use of the

machinery.

Similarly, for agricultural, forestry and residential machinery we used corresponding land use data, which is tier 1 in the guidebook. Tier 3 suggests detailed fuel deliveries with employment data for forestry and agriculture, and detailed fuel deliveries with population density and/or household numbers and types for residential. Fuel deliveries could possibly be found on e.g. regional level, possibly enhancing the spatial distribution between regions. Increasing the detail beyond land use on

forestry and agriculture is difficult. From air quality point of view these sources are not the most important, as, especially in the Nordic countries, the activities mostly take place far away from of population centres (with possible exception the more densely populated Denmark). Therefore, the land use as proxy is sufficient to get the emissions away from cities.

Roads were used very differently in the countries as proxy for machinery. Finland weighted them most, allocating road maintenance to them, while Norway didn't use road information for machinery. Denmark had much weight on the roads like

Finland, but had also high emissions on a few highways due to their construction. This highlights the differences in the spatial distribution methods between the Nordic countries, and how there is still room for development and learning from each other.



In machinery and off-road sector, CAMS-REGv4.2 inventories spatial distribution was driven mainly by population, and secondarily by agriculture land. In the Nordic inventory, only Finland used population as proxy data, and only for part of the other machinery subsector. Furthermore, the Nordic inventory utilized the variety of other proxy data more than the CAMS-REGv4.2 inventory, distributing more emissions to other areas than city centres and fields. Despite the different weighting most emissions were in same areas in all countries, except Denmark, in the two inventories. Therefore, the main characteristics of the sector were captured on the spatial resolution that the CAMS-REGv4.2 inventory had. On the Nordic inventory's higher spatial resolution more details can be seen within the high emission areas, which would impact health impact assessment results.

### 4.3. Other factors relating to spatial distribution

While the countries use the same sector nomenclature, subsectors for which the proxies have been applied are not necessarily the same. Especially the machinery and off-road sector can be divided into different subsectors, depending on the data available for activity and suitable proxies. For example, emissions can be calculated for excavators, and those emissions divided to several subsectors such as construction and industry, with specific proxies. But this is only one way to do the aggregation, and available proxies dictate the possibilities.

The impact of resolution on exposure assessment has been presented in Korhonen et al. (2019). They found that for Finland, going from 5 km to 1 km in assessment resolution had significant impact on the exposure results in urban areas. For road transport and machinery, the lower resolution of 5 km in the exposure modelling led to about 25-30% smaller mortality estimate in urban areas compared to the 1 km resolution. Decrease from 1 km to 10 km resolution caused over 40% smaller mortality estimate. In rural areas the same resolution changes from 1 to 5 km and 1 to 10 km caused 10% and 20% drop in the mortality estimate, respectively. Therefore, even though the Nordic and CAMS-REGv4.2 inventories give similar results on the CAMS-REGv4.2's resolution (5.5 km × 3.5-6.5 km, with the north-south length being shorter in north), the Nordic inventory allows more detailed health impact assessments with its higher resolution.

One aspect to consider regarding the time series of the Nordic inventory is that Finland and Norway have unchanging proxies between the years, i.e., the same spatial distribution is used for all years for a specific sub-sector. In Denmark some proxies are updated annually, and subsectors can have year-specific proxies. Sweden does the same for proxies for national to municipal/county level. Year-specific proxies can reduce the uncertainty of the spatial distribution, but they are also dependent on appropriate data. The spatial distributions don't necessarily change drastically from year to year, making static proxies more justified. However, as our inventory spans from 1990 to 2014, it is reasonable to assume that there have been changes which would affect the results in significant way. Furthermore, although out of scope of our inventory in this paper, the COVID pandemic clearly has affected the activities of people, meaning more attention needs to be given to dynamic proxies, as has been reported in Guevara et al (2022).

Uncertainties associated with the proxies were not analysed in this work. Some assessments have previously been made. Plejdrup et al. (2021) describes the assessed quality of a dataset (A to E, A being the highest) and applicability as proxy (1-5, 1 being the highest) for proxies used in Denmark, both with a five-step classification. For road transport, they assess their proxies as low uncertainty in the dataset (B, second best certainty class) and as "Good correlated proxy" (2, second best applicability class), as vast amount of traffic data is used for the proxy. For machinery and off-road the assessed ratings are presented in Table 7. In general, the quality of the datasets is considered to be better than the applicability. Main problem in applicability is thought to be lack of spatial activity data to accompany other, e.g. land use, data. Therefore, according to the report, the datasets used for proxy development have medium to low uncertainty, and the main development need is finding appropriate data to describe the activities spatially within the identified areas, for example land use classes.





**Table 7 Assessed quality of dataset used and its applicability as a proxy for machinery and off-road sector proxies in Denmark (Plejdrup et al., 2021). Quality ranges from A-E (A being the highest), and applicability from 1-5 (1 being the highest).**

| Subsector | Aviation | Railways | Building & construction | Commercial & Institutional | Forestry | Military | Residential | Agricultural |
|---|---|---|---|---|---|---|---|---|
| Quality | A | B | B | C | C | A-B | C | A |
| Applicability | 2 | 4 | 3 | 4 | 3 | 3-5 | 3 | 3 |


The Nordic emission inventory has been used as input to the coupled DEHM/UBM air pollution model system setup for the continental Nordic area within the same spatial 1 km x 1 km grid. A decadal long simulation has been evaluated against observations representing both rural and urban conditions showing that the model capture the main features in terms of spatial-temporal variability of air pollution across the Nordic (Frohn et al., 2022). This simulation has also been used for a study

comparing different health assessment tools (Lehtomäki et al., 2020) and for several health studies based on detailed Nordic health data. The studies confirm that even low levels of air pollution can be associated with negative impacts on the human health (Olsson et al., 2021; Raaschou-Nielsen et al., 2023; Sommar et al., 2021).

## 5. Data availability

The Nordic emission inventory (Paunu et al., 2023) is published as open data Zenodo, and can be accessed at
https://zenodo.org/record/8070155, https://doi.org/10.5281/ZENODO.8070155.

## 6. Conclusions

In this paper we have presented our novel Nordic air pollution emission inventory and described how spatial distribution of road transport and machinery and off-road mobile source emissions is affected by the choice of proxy data. Our inventory contains annual air pollution emissions on a high, 1 km × 1 km, spatial resolution from 1990-2014. The inventory is published

as open data to be freely used in, e.g., impact assessments (Paunu et al., 2023).

For road transport, all countries in our Nordic inventory complied with the tier 3 recommendation of the EMEP/EEA guidebook on spatial distribution of the emissions. There were different ways to categorize vehicle classes, and more detailed division could be applied in the countries, but most important was the distinction between heavy and light duty vehicles. Improvement possibilities were in replacing modelled traffic volume data with measured data where possible. Furthermore, traffic dust

emissions didn't have distinct proxies. Regional proxies, which would consider local (especially winter and spring) weather conditions, would be possible to develop and apply within each country. The issue is most relevant in Northern Europe, and, therefore, a general EMEP/EEA guidebook update might not be needed.

Comparison to the European wide CAMS-REGv4.2 inventory showed that their proxy, based more on modelled traffic volumes, compared to the proxy in the Nordic inventory, based on mix of locally measured and modelled traffic volumes,

created lower emission weights on highways and capitals, and higher weights in other cities and population centres. In other words, modelled traffic volumes based on population didn't catch the extent of traffic on larger roads. This would have impact on population exposure and health impact assessments, as emissions close to population will increase the estimated impacts. Therefore, more attention should be focused on traffic volume modelling to improve the spatial proxies for road transport.

The proxies for machinery and off-road emissions in our Nordic inventory differed from the tier 3 recommendations in the

EMEP/EEA guidebook. The guidebook recommendations for machinery and off-road are included within broader sectors, and suggest point source and fuel delivery data to be used. For many subsectors, our Nordic inventory utilizes land use data, which is tier 1 (most simple tier) in the guidebook. However, we argue that for many machinery and off-road activities the land use data is the best available starting point for the proxies, as explained in the Discussion section. From health impact assessment point of view, for many emission sectors, such as agriculture and forestry, the main issue is to distribute the emissions away



from population centres, which is easily achieved with suitable land use classes. The distribution can be enhanced by using available data related to the activity, such as regional tractor distribution or annual loggings, or the suggested fuel deliveries. CAMS-REGv4.2, despite having less complex proxies for machinery and off-road, was able to produce similar spatial patterns for the emissions as our Nordic inventory in three of the four countries. The conclusion holds for the resolution of the CAMS-REGv4.2 inventory. Our Nordic inventory has higher native resolution (1 km × 1 km vs. 0.1° × 0.05° of CAMS-REGv4.2)

and offers more details in the spatial pattern. This is especially important in cities, where both population density and emissions are highest. Therefore, the Nordic inventory allows for more detailed health impact assessments.

We suggest that the EMEP/EEA guidebook should have separate recommendations for the machinery and off-road proxies instead of including them within more broad sector division. Furthermore, measured traffic flows should be utilized where possible in lieu of modelled data. Land use data is a good starting point for a proxy, and can with success be combined with

other activity specific data to enhance the quality of the spatial distribution.

## Author contribution

Ville-Veikko Paunu: Conceptualization, Methodology, Formal analysis, Writing - Original Draft, Visualization. Niko Karvosenoja: Conceptualization, Methodology, Writing - Original Draft, Supervision, Project administration, Funding acquisition. David Segersson: Methodology, Writing - Review & Editing. Susana López-Aparicio: Methodology, Writing -

Review & Editing. Ole-Kenneth Nielsen: Methodology, Writing - Review & Editing. Marlene Schmidt Plejdrup: Methodology, Writing - Review & Editing. Throstur Thorsteinsson: Methodology, Writing - Review & Editing. Dam Thanh Vo: Methodology, Writing - Review & Editing. Jeroen Kuenen: Methodology, Writing - Review & Editing. Hugo A.C. Denier van der Gon: Methodology, Writing - Review & Editing. Jukka-Pekka Jalkanen: Methodology, Writing - Review & Editing. Jørgen Brandt: Project administration, Funding acquisition, Writing - Review & Editing. Camilla Geels: Project

administration, Funding acquisition, Writing - Review & Editing.

## Competing interests

The authors declare that they have no conflict of interest.

## Acknowledgements

This research was funded by: Nordforsk under the Nordic Programme on Health and Welfare (Project #75007: Understanding

the link between air pollution and distribution of related health impacts and welfare in the Nordic countries, NordicWelfAir); SIS-EMISYS project "Emission system for climate gases, air pollutants and other environmental contaminants" (B120011); Ministry of the Environment, Finland; the project BioDiv-Support project funded though the 2017-2018 Belmont Forum and BiodivERsA joint call for research proposals, under the BiodivScen ERA-Net COFUND programme, and with the funding organisations AKA (Academy of Finland contract no 326328), ANR (ANR-18-EBI4-0007), BMBF (KFZ: 01LC1810A),

FORMAS (contract no:s 2018-02434, 2018-02436, 2018-02437, 2018-02438) and MICINN (through APCIN: PCI2018-093149).

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
