# Peer review of "Air pollution emission inventory using national high-resolution spatial parameters for the Nordic countries, and analysis of PM2.5 spatial distribution for road transport and machinery & off-road"

_Earth System Science Data, 2023_

## Author Response (AR1)

**Response to Reviewers**

**Air pollution emission inventory using national high-resolution spatial parameters for the Nordic countries, and analysis of PM$_{2.5}$ spatial distribution for road transport and machinery & off-road**

Ville-Veikko Paunu[1], Niko Karvosenoja[1], David Segersson[2], Susana López-Aparicio[3], Ole-Kenneth Nielsen[4], Marlene Schmidt Plejdrup[4], Throstur Thorsteinsson[5], Dam Thanh Vo[3], Jeroen Kuenen[6], Hugo Denier van der Gon[6], Jukka-Pekka Jalkanen[7], Jørgen Brandt[4,8], Camilla Geels[4]

We thank Associate Editor Dr. Yuqiang Zhang for considering our article and the reviewers for great comments which help to clarify our main messages and improve the readability of our manuscript. We have addressed all comments and made appropriate edits and responses to them.

**Reviewer #1:**

> This manuscript provides high-resolution air pollutant emission data for Nordic countries from 1990, 1995, 2000, 2005, 2010, 2012, and 2014. This data is worth publishing on ESSD; however, I have some concerns about the data description in this manuscript. The manuscript discusses and evaluates the primary PM2.5 emission contribution from emission sectors and Nordic countries. Other species (such as SOx, NOx, CO, NMVOC, NH3), also present in the data, are only discussed in the time series in Figures 1 and 2. I suggest adding more information for other species (NOx, CO, NMHC, and NH3) regarding the contribution of sectors and countries, spatial distribution discussion, and a comparison with the other emission inventory.

> Specific Comments:

> 1. Line 231: What is "SIMAIR"?

> 2. In Table 3, why is Finland's air traffic (Aviation) PM2.5 emission zero in the table, but the gridded emission inventory data on Zenodo shows Finland's air traffic emission data (variable 12, starting from 0)?

> 3. What is the emission year for Table 3 (Line 400)?

> 4. In Table 3, it shows off-road emission. I suggest adding one row at the top for on-road emissions for all Nordic countries. The table should summarize all sectors' contributions and countries.

> 5. Lines 379 to 388: The description "Spatial distribution of … can be expected" pertains to method details. Lines 389 to 392 only present some results and should add more details, such as the largest emission cities and the country with their emission rates for individual species. I suggest re-editing this section.

> 6. Sections 3.1 and 3.2 only discuss the primary PM2.5 emission and contribution; other species in the inventory data are not discussed in their spatial distribution.

> 7. Line 425: "Figure 5 Machinery (SNAP 8) PM2.5…" I think it should be "Machinery

and off-road"?

8. Table 6: PM2.5 emission in 2010 for Machinery off-road part differs from Table 3's Machinery off-road total; are those two tables from different years? Please confirm the details.

9. Figures 7 and 10 show the TNO inventory on the X-axis, but the caption did not specify the TNO inventory (I think it is the CAMS-REGv4.2 inventory). Please use a consistent name for the emission inventory. Furthermore, the X and Y axes are better at the same scale and keep the dashed yellow as the diagonal line.

10. The data separates the gridded shipping emission from other gridded emissions, but there are still empty variables for other emission sectors in the Shipping emission file (variables 1 to 19). Combining them into one file or removing the empty variables is better.

**Authors' response:**

We thank the reviewer for the excellent comments which help to clarify and enhance our manuscript. To respond to the comment on the other emission species and sectoral and country contributions, we will add a table of all emissions per country and per sector to the supplement, both for our inventory and the CAMS-REGv4.2. To respond to the comment on spatial distributions, we will also add emission maps of all the emission species for our inventory to the supplement.

Specific comments:
1. SIMAIR is a tool for assessing emissions, e.g. PM10, for several sectors, including non-exhaust emissions from road transport. We will include the description of the tool and the following reference to the methods: https://doi.org/10.1016/j.atmosenv.2008.01.056.
2. The official inventory that our Nordic inventory is based on doesn't have air traffic PM2.5 emissions for Finland for 2012 and 2014 but has it for the other years. This is a mistake in the inventory, but as the emissions would be very small even if included, the omission of these emissions don't affect modelling results in a significant way. The subsector only considers landing and take-off emissions. This omission will be fixed in future updates of the dataset.
3. The year for Table 3 is 2014. We will add the missing information to the caption.
4. We will add the on-road transport emissions to table 3 as suggested. Emissions for the other sectors are added to the supplement.
5. We will re-edit the section to respond to the points raised. The first part of the section does focus on the methods, as rightly pointed out by the reviewer. As the one main result of the study is to identify the ways to spatially distribute emissions for the two sectors in question, and to identify the differences between the countries, we defend the text inclusion in the result section. Road transport sector has established methods for the spatial proxies, but even so there are differences in the details how each country handles the important characteristics of the sector, as well as more minor specifics (such as Sweden's weighting emissions to areas outside roads). We will add a sentence to highlight this important point. To respond to the comment on the cities, we will add text on how the capitals had highest emission weights in each country, although Norway had high emissions in other larger cities as well. Norway also weighted less emissions to highways compared to other countries. We will elaborate these differences more the section. We will also include maps and emission tables of other pollutants to the supplementary.

6. This is an important point. As the spatial proxies are sector and not pollutant specific, the same conclusions we make based on PM2.5 emissions hold for other pollutants as well. Furthermore, the dataset is mainly developed for health impact assessment, and PM2.5 is the main pollutant for health impacts. Furthermore, primary PM2.5 has largest impact to local air quality, and, therefore, the high-resolution spatial assessment of PM2.5 emissions has the most significance for health assessments. For these reasons we focus on PM2.5. We will make this clearer in the discussion section. We will also add emission maps for the other pollutants to the supplementary.
7. Comment is correct, the caption should be "Machinery and off-road…". We will add the missing text to the caption.
8. The year for the emissions in Table 6 is indeed 2010, and for Table 3 the year is 2014. We will add the year to Table 3 to clear this up.
9. We will update the labels on the Figures 7 and 10 to specify the inventory, and update the scales on the X and Y axes as suggested to improve the graphs.
10. This is a good point. We will update the dataset so that shipping emission file will contain only the shipping sector, and the main file will not include the empty shipping sector. We will keep the shipping in separate file as it is based on separate data, and we encourage future users to look for most up to date shipping emissions if needed.

**Reviewer #2:**

This study constructed the Nordic air pollution emission inventories covering 1990, 1995, 2000, 2005, 2010, 2012, and 2014. It highlighted the differences in the spatial distribution of emissions from road transport, machinery and off -road sectors from the CAMS-REGv4.2. On this basis, suggestions for changes to the future EMEP/EEA guidelines are presented. Overall, this is a good work, but more effort may be needed in terms of refining the content of the article.

Main concerns:

1. Is it possible to add some simulations or observations to confirm that the list is more accurate than CAMS-REGv4.2?
2. It may be possible to add more analytical information, such as the spatial distribution of different species, changes in spatial distribution under different time series, changes in the contribution of emissions from different countries or sectors, etc.

Other minor concerns include:

1. The title of the article is "Air pollution emission inventory using national high-resolution spatial parameters for the Nordic countries", but the focus is on road transport and machinery. The authors need to consider whether to adjust the title of the article.
2. Legend of the map should be shown as a range of values rather than a single value.
3. Some abbreviations should be used in full when they first appear, e.g. SIMAIR, SNAP.
4. Figure 7: What is the TNO inventory, is it CAMS-REGv4.2?
5. Is the overall low level of emissions from the Nordic inventories in Figure 7 supported by additional and more detailed evidence?

**Authors' response:**

We thank the reviewer for the good and helpful comments which help to clarify our main messages and improve the manuscript.

Main concerns:
1. The Nordic inventory has been implemented into the DEHM/UBM model system going to a resolution of 1 km x 1 km across the Nordic area (Denmark, Finland, Norway and Sweden) using a Eulerian-Gaussian modelling approach. The model system has been run for the period 1979-2018 providing hourly concentrations of the main air pollutants (like O3, NO2, PM2.5). These data have been evaluated against observations representing rural and urban background conditions in the four countries (Frohn et al., 2022, doi.org/10.1016/j.atmosenv.2022.119334). The comparison showed that the DEHM/UBM model system performs well in general; the annual, country-level correlation coefficients were in the range from 0.7-0.9 for NO2, 0.6-0.7 for O3 and 0.4-0.9 for PM2.5. This is discussed in the discussion section, and we will add more details there to demonstrate how the data has performed compared to measured air pollution levels. While there are no direct comparisons to how the CAMS-REGv4.2 compares to these results, we have discussed how our Nordic inventory allows for higher spatial resolution for modelling, which is especially important in cities with higher population density. Our inventory has been developed and collected by local national experts, and is able to take national characteristics better into account compared to larger scale inventories, such as the CAMS-REGv4.2, which uses common methods over larger areas. In addition to national spatial data, our Nordic inventory applies national emission factors instead of common, European level ones, as described in Paunu et al. (2021) (doi.org/10.1016/j.atmosenv.2021.118712).
2. These are all important points. As written in the answer to RC1, question 6, the proxies are sector-specific, and not pollutant dependent, and, therefore, the subsectoral analysis holds for other pollutants than PM2.5. We will make this clearer in the article, and we will add emission maps for all pollutants to the supplementary. Time series question is addressed in the discussion, i.e. for most sectors and countries the proxies are static, i.e. they are the same for all years. Denmark and Sweden do have some proxies changing from year to year. We will emphasis this more by including the main points to the conclusions. To demonstrate the contribution of the countries and sectors between the years, we will include tables of the emissions to the supplementary.

Minor concerns:
1. Thank you for the good point. We will adjust the title to "Air pollution emission inventory using national high-resolution spatial parameters for the Nordic countries, and analysis of PM2.5 spatial distribution for road transport and machinery & off-road".
2. The legend shows the colour of the emission value at the exact value. Between the values, the colours are interpolated between the ones shown. We will update all the captions for the maps to make this clearer.
3. We will explain the abbreviations when first used for SIMAIR, SNAP, STEAM, and NORTRIP.
4. The TNO inventory is CAMS-REGv4.2. We will update the labels to clarify this.
5. As responded to the 1st main concern of the reviewer, we have discussed comparisons of modelled pollution concentrations in the discussion section, and will elaborate the part with more detailed results from these comparison. They show that in general the modelled results perform well when compared to measured concentrations.